

# Contrasting juxtaposition of two paradigms for diazotrophy in an Earth System Model of intermediate complexity

Ulrike  Löptien[1,2] and Heiner Dietze[1,2]

[1]GEOMAR Helmholtz Centre for Ocean Research Kiel, Düsternbrooker Weg 20 24105 Kiel, Germany
[2]Institute for Geosciences, University of Kiel, Ludewig-Meyn-Str. 10, 24118 Kiel, Germany
**Correspondence:** Ulrike Löptien (uloeptien@geomar.de)

**Abstract.** Nitrogen fixers, or diazotrophs, play a key role in the carbon and nitrogen cycle of the world oceans, but the controlling mechanisms are not comprehensively understood yet. The present study compares two paradigms on the ecological niche of diazotrophs in an Earth System Model (ESM). In our standard model configuration, which is representative for most of the state-of-the-art pelagic ecosystem models, diazotrophs take advantage of zooplankton featuring a lower food preference for

diazotrophs than for ordinary phytoplankton. We compare this paradigm with the idea that diazotrophs are more competitive under oligotrophic conditions, characterized by low (dissolved, particulate, organic and inorganic) phosphorous availability. Both paradigms are supported by observational evidence and lead to a similar good agreement to the most recent and advanced observation-based nitrogen fixation estimate in our ESM framework. Further, we illustrate that the similarity between the two paradigms breaks in a RCP 8.5 anthropogenic emission scenario. We conclude that a more advanced understanding of the

ecological niche of diazotrophs is mandatory for assessing the cycling of essential nutrients, especially under changing environmental conditions. Our results call for more *in-situ* measurements of cyanobacteria biomass if major controls of nitrogen fixation in the oceans are to be dissected.

## 1   Introduction

Nitrogen is an essential element in the metabolism of plants. Even though it is much more abundant in air than for example

carbon dioxide, the assimilation of carbon by phytoplankton in vast regions of the ocean is considered to be limited by the availability of nitrogen. Among the reasons for this is that most phytoplankton cannot use the molecular nitrogen, that is so abundant in air, because it cannot break the exceptionally strong chemical bond between the two nitrogen atoms constituting molecular nitrogen ($N_2$). An exception to this rule are so-called nitrogen fixing microorganisms (bacteria and archaea) that are capable of breaking this bond and thus can utilize the molecular nitrogen of the air. Their total input of bioavailable

nitrogen to the ocean is substantial: current estimates range from 70 to 200 $TgNyr^{-1}$ (c.f., Großkopf et al., 2012; Gruber and Sarmiento, 1997; Tang et al., 2019b; Wang et al., 2019). Paleo-records suggest that for the last several thousand years, this input has been balanced by denitrification and anaerobic ammonium oxidation (anammox) which converts bioavailable forms of N (e.g. $NO_3^-$, $NO_2^-$) under low-oxygen conditions back to $N_2$ (Altabet et al., 2012). The apparent balance of sources and sinks of bioavailable N suggests a coupling mechanism between these sources and sinks. Such coupling mechanisms and





related controls on the overall oceanic N-budget are discussed controversially (e.g., Gruber, 2016; Wang et al., 2019) - also because the controls on nitrogen fixation are not comprehensively understood.

There is, however, consensus, that anthropogenic forcing, such as global warming and input of bioavailable nitrogen to the ocean, modulates the turnover of bioavailable nitrogen in the ocean which might have far-reaching consequences (Duce et al., 2008; Krishnamurthy et al., 2007; Krishnamurthy. et al., 2009; McMahon et al., 2015). Some studies suspect that $N_2$-

fixation might increase regionally in the future, adding extra nutrients to already-overfertilized systems (e.g., Tang et al., 2019; Wasmund et al., 2001), or even result in vicious cycles (e.g., Vahtera et al., 2007). This comprehension has triggered efforts to capture the respective dynamics in numerical models which explicitly resolve oceanic transport mechanisms (such as currents and mixing) along with the turnover of biogeochemical species (nutrients, carbon and oxygen) - with the aim to forecast the effects of changing environmental conditions and thereby facilitating management and mitigation measures.

A generic approach to capture the major aspects of the marine N-cycle in numerical models is to include an explicit representation of diazotrophs as an extra functional group, that is distinctly different to the representation of ordinary non-fixing phytoplankton (e.g., Fennel et al., 2001; Moore et al., 2001; Paulsen et al., 2017). The underlying model assumptions and respective mathematical formulations vary from one model to another, but they typically agree on (1) diazotrophs are capable of utilizing $N_2$, while ordinary phytoplankton is not and (2) diazotrophs grow slower (or at least not faster) than ordinary phyto-

plankton, because of the metabolic cost involved in providing means to break the strong bond between the two nitrogen atoms constituting $N_2$. When taken alone, a direct deduction of these two assumptions is that diazotrophs only have an advantage over ordinary phytoplankton in regions where bioavailable nitrogen supply is below the demands of ordinary phytoplankton, relative to the phosphorus supply (Deutsch et al., 2007; Schmittner et al., 2008). In the remainder of the ocean, ordinary phytoplankton would suppress the relatively slow growing diazotrophs by competing more successfully for resources, such as bioavailable

phosphorus (P), iron (for which diazotrophs have a higher demand) and light (e.g., Falcón et al., 2005; LaRoche and Breitbarth, 2005; Mills and Arrigo, 2005). Observations, however, indicate no such correlation between nitrogen fixation and a nutrient supply that is low in nitrogen relative to, e.g., bioavailable phosphate (Capone et al., 2005; Gruber, 2004). Further perplexity is added by the fact that diazotrophs are known to increase their internal N:P ratios via fixation beyond the needs of ordinary phytoplankton. Thereby they ultimately overstock the system with extra nitrogen - which would downsize the extent of their

very own ecological niche (c.f., Landolfi et al., 2013).

All prognostic models we have reviewed so far, include some additional assumptions that give diazotrophs an additional competitive advantage, that outweighs their tendency to change their environmental conditions to their disadvantage (by adding extra nitrogen to the system). These additional assumptions are not always explicitly formulated, but are often imposed during trial and error exercises where model parameters are tuned in an attempt to get a good fit to observations. One assumption often

implicitly incorporated (but rarely explicitly mentioned) is *selective grazing* by zooplankton, where ordinary phytoplankton is exposed to higher grazing pressure than the (slower-growing) diazotrophs (e.g., Keller et al., 2012; Paulsen et al., 2017). This paradigm may be founded on the fact that some diazotrophs can produce toxins that fend off grazers (Hawser et al., 1992). An alternative paradigm, equally consistent with observational studies (e.g., Degerholm et al., 2006), has been suggested recently by Pahlow et al. (2013) who use an optimality based model to illustrate, that oligotrophy - especially in P, can favour





diazotrophy. The idea is that diazotrophs outcompete ordinary phytoplankton under P-depleted conditions because they can allocate more P to intracellular P-uptake machinery. The underlying physiological explanation is that diazotrophs presumably need less P to build their N-uptake machinery, which is already taken care of by the nitrogen-fixation machinery (which is known not to need so much P). Flavors of this paradigm, *low P-demands* of diazotrophs, have been explored in the literature already (Mulholland et al., 2002; Landolfi et al., 2015). In the present study, we add to the ongoing discussion by comparing

the implications of the *selective grazing* with the *low P-demands* paradigm. A major focus is on future projections, carried out with the UVic 2.9 Earth System Model (Weaver et al., 2001). The major aim is to illustrate the far reaching implications of both paradigms in a generic model that is currently used to project into our warming future and to assess geoengineering options (see, e.g., Keller et al., 2014; Kemena et al., 2019; Mengis et al., 2016; Reith et al., 2016). Our experiments are designed to constrain the envelope of model responses as set by each of the paradigms. We find (and illustrate in this study) that the

development of a reliable model must be preceded by additional in-situ observations.

## 2  Methods

In the following, we describe the Keller et al. (2012) reference model (Sect. 2.1) and two flavors of this reference model (Sect. 2.2), representative of the *selective grazing* and *low P-demands* paradigm, respectively. Sect. 2.3 describes the numerical experiments and Sect. 2.4 the considered observations.

### 2.1  The Reference Model

We use the UVic 2.9 Earth System Model, documented and extensively assessed in Keller et al. (2012). We refer to this standard model version since it is extensively used (e.g., Keller et al., 2014; Kemena et al., 2019; Reith et al., 2016), well documented (Keller et al., 2012; Schmittner et al., 2008) and was used earlier to study diazotrophy (e.g., Landolfi et al. (2017); Somes and Oschlies (2015)). The model is of intermediate complexity and thus allows for running a multitude of simulations

to quasi-equilibrium. The horizontal resolution is $1.8°$ in latitude and $3.6°$ in longitude in all submodules (ocean, including biogeochemistry, dynamic-thermodynamic sea ice module, atmospheric energy-moisture balance model, land module, with an active terrestrial vegetation component, and simple land ice). The ocean submodule is based on a three-dimensional primitive-equation model (Pacanowski, 1995). The vertical discretization of the ocean comprises 19 levels and increases gradually from $50\,m$ at the surface to $500\,m$ at depth. The vertical background mixing parameter, $\kappa_h$, is $0.15\,cm^2s^{-1}$ throughout the water

column. In the Southern Ocean (south of $40°$S) this background value, $\kappa_h$, is increased by $1.0\,cm^2s^{-1}$. An anisotropic viscosity scheme, suggested by Large et al. (2001), is implemented to improve the equatorial circulation (Somes et al., 2010). A marine pelagic biogeochemical model is coupled to the ocean circulation component. Its prognostic variables are ordinary (i.e. non-diazotrophic) phytoplankton ($P_o$), diazotrophic phytoplankton ($P_d$), zooplankton (Z), detritus (D), nitrate ($NO_3$), phosphate ($PO_4$), dissolved oxygen ($O_2$), dissolved inorganic carbon (DIN) and alkalinity (ALK). The model is described in detail in

Keller et al. (2012) and Schmittner et al. (2008). Note that the dispersive numerics of the oceanic advection occasionally




produces small negative values in phytoplankton and zooplankton concentrations. These values are set to zero in the model evaluation.

In the following, we present a choice of details relevant for the simulated dynamics of diazotrophs (relevant model parameters are listed in Tab.1): phytoplankton growth is generally controlled by the availability of light and nutrients (here, nitrate, phosphate and iron, where the latter is parameterized rather than explicitly resolved). Phytoplankton blooms are terminated by zooplankton grazing, once the essential nutrients are depleted and the related reduction in phytoplankton growth does not longer keep up with the grazing pressure, that has built up during the bloom. For ordinary phytoplankton the maximum potential growth rate is

$$J_o^{max} = a \cdot \left(\frac{Fe}{k_{Fe} + Fe}\right) exp(T/15.65),$$ (1)

where $k_{Fe}$=0.1 $mmol\ Fe/m^3$ is the *half-saturation constant* for iron (Fe)-limitation and $a = 0.6\ d^{-1}$ determines the maximum potential growth rate at an ambient temperature $T = 0°C$.

For diazotrophs the formulation of the maximum potential growth is similar, but diazotrophs are disadvantaged in nitrate-replete waters by $C_d = 0.4$. In addition, growth is stalled in waters colder than $15°C$:

$$J_D^{max} = C_d \cdot a \cdot max\left(0, \frac{Fe}{k_{Fe} + Fe}\right) \cdot exp((T/T_b) - 2.6).$$ (2)

The actual growth rate of non-diazotrophic phytoplankton, $J_O$, is, in case of low irradiance (I) and/or nutrient depleted conditions, the maximum potential rate $J_o^{max}$ reduced by the following implementation of Liebig's law of the minimum:

$$J_O = min\left(J_{IO}, J_O^{max} \frac{NO_3}{k_N + NO_3}, J_O^{max} \frac{PO_4}{k_P + PO_4}\right),$$ (3)

where $k_N$ and $k_P$ are the *half saturation constants* for $NO_3$ and $PO_4$, respectively.

The actual growth rate of diazotrophs, $J_D$, is similar but it is not affected by nitrate deficiency:

$$J_D = min\left(J_{ID}, J_D^{max} \frac{PO_4}{k_P^d + PO_4}\right),$$ (4)

The light-limited growth $J_{IO}$ and $J_{ID}$ are both determined by:

$$J_{IOandD} = \frac{\alpha I J_{OandD}^{max}}{(\alpha I)^2 + (J_{OandD}^{max})^2}.$$ (5)

The initial slope of the P-I curve ($\alpha$) is set to 0.1 $(W/m^2)^{-1}d^{-1}$. Diazotrophs take up available nitrogen following:

$$NO3_{upt} = min\left(J_{ID}, \frac{min(NO_3, PO_4 \cdot 16)}{(k_P^d \cdot 16) + min(NO_3, PO_4 \cdot 16)} \cdot J_D^{max}\right) \cdot P_D,$$ (6)

where $P_D$ denotes the biomass of diazotrophs (in units $mol N/m^3$). When no or not enough bioavailable N is available, nitrogen fixation tops the respective N pool up to a Redfield-ratio of N:P=16. Note that the above formulation differs slightly from the original formulation of Keller et al. (2012) which reads:

$$NO3_{upt} = min\left(J_{ID}, \frac{PO_4}{k_P^d + PO_4} \cdot J_D^{max}\right) \cdot \frac{NO_3}{(k_P^d \cdot 16) + NO_3} \cdot P_D,$$ (7)

Our minor change ensures a more realistic behavior where no nitrogen is fixed under nitrate-replete conditions. When applied to the reference model version, the difference to the original Keller et al. (2012) simulation turned out to be negligible.





The reference version of Keller et al. (2012) is an implementation of the preferential grazing paradigm. Grazing is determined
by:

$$Graze = \mu_{max} \cdot Z \cdot \theta \cdot P, \tag{8}$$

with a maximum growth rate $\mu_{max} = 0.4$ d$^{-1}$. Grazing on ordinary phytoplankton, $P_O$, is calculated by setting $\theta = \theta_o = 0.3$.
Grazing on diazotrophs, $P_D$, is calculated with a lower grazing preference $\theta = \theta_d = 0.1$.

## 2.2 Implementation of the two Paradigms

We integrate two sets of numerical model configurations. The first set explores the paradigm *selective grazing*, represented by
the model setup GRAZ and assumes that diazotrophs are grazed less than ordinary phytoplankton. The second set explores
the paradigm *low P-demands*, represented by the model setup OLIGO. We consider both paradigms individually to constrain
the envelope of model responses as set by each of the paradigms. To ensure comparability between the paradigms we identify
parameters for both paradigms which represent available observations in an "optimal" way (see below). Finally we compare
the respective "optimal" representatives of the two paradigms, OLIGO and GRAZ, in Section 3.

The *selective grazing* paradigm is implemented by adjusting the grazing preferences for diazotrophs and ordinary phyto-
plankton, $\theta_d$ and $\theta_o$, with the additional constraint $\theta_d + \theta_o = 0.4$. In addition, $C_d$ is varied in attempt to minimize the misfit
to observations. In total, we explored 24 combinations of the parameters $\theta_d$, $\theta_o$ and $C_d$. In the following, GRAZ refers to the
most realistic member of the set of 24 that outperforms the reference version of Keller et al. (2012) (c.f., Tab. 1). The respective
model assessment metric is based on the most recent estimate for global nitrogen fixation, provided by Wang et al. (2019).
More specifically we calculate the sum of absolute deviations between the locally estimated nitrogen fixation of Wang et al.
(2019) and our simulated nitrogen fixation (mean absolute error):

$$cost = \left| \left( \frac{1}{N_m} \sum_{j=1}^{N_m} (a_j - \hat{a}_j) \right) \right|. \tag{9}$$

$N_m$ denotes the number of estimated values for nitrogen fixation (on the model grid), $a_j$ stands for an 'observation' at location
$j$ and $\hat{a}_j$ for the corresponding model result. We consider climatological annual means. Note that this choice of misfit metrics
inevitably adds a subjective element and the respective choice will impact the results of the optimization procedure (see Löptien
and Dietze, 2015, 2017). A choice has to be made, however, and here we follow a standard approach (c.f., Pontius et al., 2008;
Sauerland et al., 2018; Willmott and Matsuura, 2005). Note that solutions with very low biomass of diazotrophs (below an
integrated value of 10 $TgC$) are not considered.

The second paradigm, represented by OLIGO, assumes that diazotrophs can cope better with oligotrophic conditions than
ordinary phytoplankton. This paradigm is implemented by (1) setting equal grazing preferences for diazotrophs and ordinary
phytoplankton conforming with the additional constraint of $\theta_d + \theta_o = 0.4$ and (2) tuning the half-saturation constant for phos-
phate limitation of diazotrophs $k_P^d$ (where the values considered are smaller than the original value). Again, $C_d$ is varied in
addition to minimize the misfit to observations. In total we explore 24 values for $k_P^d$ and $C_d$. The respective parameter ranges

are listed in Tab.1. In the following OLIGO refers to the most realistic member of the set of 24 that, according to the metric defined in Equation 9, outperforms the reference version of Keller et al. (2012) (c.f., Tab. 1). Please note that in terms of misfit

to observed climatologies of phosphate, nitrate and oxygen REF, GRAZ and OLIGO perform comparable to one another: when considering the World Ocean Atlas 2005 (Garacia et al., 2006), all respective differences in the (volume weighted) root mean squared errors of the three model versions are smaller than the corresponding average (volume weighted) standard deviations of the observations.

## 2.3 Numerical Experiments

The two sets of tuning experiments start from the equilibrated state of what is dubbed *reference run* in Getzlaff and Dietze (2013) and are integrated for 2000 model years with constant, pre-industrial $CO_2$ emissions. Respective experiment evaluations (as carried out with Equation 9) refer to the average of the last 10 years of these simulations, hindcasting historic conditions. In addition, we will show results from simulations of GRAZ and OLIGO, covering the period 1800-2150. They start with the historical state and are continued with the $CO_2$ emission scenario RCP 8.5 (Riahi et al., 2011).

## 2.4 Observational Data

*In situ* observations of diazotrophic biomass and nitrogen fixation rates, such as compiled by Luo et al. (2012), are very useful but still too sparse to allow for robust comparisons between model and observations. This has been a serious drawback hindering the identification of the major controls of diazotrophy. This study exploits the most recent and most comprehensive product from Wang et al. (2019), that fills data gaps by combining observations with inverse biogeochemical and prognostic

ocean modelling - an approach pioneered by Deutsch et al. (2007).

## 3 Results

Our tuning exercises for the two paradigms *selective grazing* and *low P-demands* identified two respective "best" simulations out of two sets of 24 model integrations, which, according to Equation 9, feature the highest similarity to the most recent estimate of global oceanic nitrogen fixation (setups GRAZ and OLIGO; c.f., Sect.2.2). In the following, we present the respec-

tive model results with respect to diazotrophy under historic climate conditions. These results are put into perspective to the reference model version REF. We continue with the results from RCP 8.5 scenario integrations.

Both "best" representatives of the two considered paradigms, GRAZ and OLIGO, feature a very similar fit to the observational estimate for nitrogen fixation, clearly superior to the reference simulation REF (the respective costs are 18.2 and 17.6, versus 27.7 $mmol\ N/m^2/yr$). A comparison of our simulations with the, unfortunately very sparse, but independent (in the

sense that this data has not been used to identify our the "best" simulations with Equation 9) data collection of diazotrophic biomass from Luo et al. (2012), indicates that the simulated distribution of diazotrophs based on OLIGO is more realistic than in GRAZ. The biomass of diazotrophs in GRAZ is apparently too low. When taking the upper 10% percentiles as a measure for bloom intensities (derived from the respective histograms of positive local annual mean values), OLIGO lies with 716





$mg\ C/m^2$ much closer to the observations (598 $mg\ C/m^2$ ) than GRAZ (282 $mg\ C/m^2$) and REF (320 $mg\ C/m^2$). Please

note, however, that the model data are climatological while the observations are rather anecdotal such that the comparison

relies on the implicit assumption that, e.g., seasonal variations are small (also assumed by, e.g., Landolfi et al., 2015).

Figure 1 and 2 give a visual impression of the above listed numerical estimates for the goodness of fit. The reference

simulation features too sharp spatial gradients in nitrogen fixation, when compared to the estimate of Wang et al. (2019)

(Fig.1a and d). Particular high values for the simulated fixation in REF occur in the tropical Pacific at the edge of the upwelling

regions and in the Indian Ocean, while the observational estimate shows a more homogenous pattern in east-west direction.

Consistent with the fact that OLIGO and GRAZ are tuned against this estimate, the spatial pattern in the simulated amount

of fixed nitrogen in both these simulations are smoother when compared to REF and the maximum values roughly match the

observational estimate.

Even though OLIGO and GRAZ are very similar, some smaller differences among the two are evident: while GRAZ features

generally rather low fixation rates in the Southern Hemisphere, this model version seems most realistic in the North Atlantic.

With respect to diazotroph biomass, the differences between the setups are more pronounced. While the simulated pattern in

nitrogen fixation roughly matches the distribution of diazotrophs in REF (Fig.1a, compared to Fig.2a) and GRAZ (Fig.1b,

compared to Fig.2b), this relation is not as evident for OLIGO: e.g., the amount of fixed nitrogen in the western tropical Pacific

is rather low compared to abundance of diazotrophs (Fig.1c, compared to Fig.2c). The major reason why the overall pattern

in dizotroph abundance and fixation do not fully match in OLIGO is that diazotrohs find an ecological niche in oligotrophic

regions (as they require less P than ordinary phytoplankton) without the need to fix atmospheric nitrogen. As desert dwellers

they occupy more area but, even so, they can not necessarily sustain high production rates since access to essential bioavailable

phosphorous is limited. Correspondingly, the quota of annual mean fixation rates versus the biomass of diazotrophs is in

OLIGO more than 3 smaller than in REF and factor 2 smaller than in GRAZ. Also, in contrast to REF and GRAZ, minimum

values even go down to zero for OLIGO. Note that these quotas were based on locations with a diazotrophic biomass of more

than 5 $mgC/m^2$ only (to avoid division by very small numbers).

In a next step, we determine the parameter sensitivities for the two paradigms, *selective grazing* and *low P-demands* (which

optimal solutions are represented by GRAZ and OLIGO, resp.). Specifically, we explore the sensitivity of changes in the half

saturation constant, $k_P^d$, versus changes in the food preference, $\theta_d$ (based on the respective ensembles of simulations described

in Sect.2.2, while $C_d$ is kept constant on the respective optimal level). Changing both parameters, $\theta_d$ and $k_P^d$, by 20% of their

covered range (0.02 and 0.0088, resp.) leads on average to a much larger change in the amount of fixed nitrogen for the *se-*

*lective grazing* paradigm, compared to *low P-demands* (16.8 and 0.8 $Tg\ N\ yr^{-1}$, respectively). For the integrated biomass

of diazotrophs however, the respective changes are similar for both paradigms (4.9 $Tg\ C$ for *selective grazing* v.s. 5.8 $Tg\ C$

for *low P-demands*). Note, in this context that for both paradigms, the biomass of diazotrophs reacts strongly nonlinear to

parametric changes. Interestingly, decreasing the food preference for diazotrophs can even lead to a *tipping point* where the

spatial distribution of diazotrophs is shifted entirely to the upwelling regions.





We conclude that the paradigm *low P-demands* produces a model behavior that is much more robust towards parameter changes than the *selective grazing* paradigm, in the sense that small changes in (rather unconstrained) model parameters effect

moderate changes in the quality of the fit to observations. In contrast, small changes to the *selective grazing* paradigm might even induce a regime shift.

A question which arises in this context, is whether the relatively large parameter sensitivity of the paradigm *selective grazing* is also reflected in the model sensitivities towards environmental changes. To explore the effects of changing climate conditions, we projected the model versions OLIGO and GRAZ into a warming future, corresponding to the RCP 8.5 emission scenario.

The response of both, OLIGO and GRAZ, are closely tied to an increasing vertical stratification effected by an ocean, that is warmed from above. The water expands at the surface and this increases the vertical density gradient, such that the energy requirements for vertical mixing are increased, because mixing is now associated with pushing lighter, more buoyant water downwards. The increased energy demands for mixing result in a dampening of mixing events and, overall, in less nutrients mixed upwards to the sun-lit surface. Among the processes setting in are (1) a reduction of phosphate supply to the surface

(globally and in oligotrophic regions), (2) an increase of oligotrophic regions where phosphate concentrations are depleted at the surface. Such an increase in the size of the oligotrophic regions is captured well in both model versions: following Polovina et al. (2008) we define oligotrophic regions as regions featuring a chlorophyll *a* concentration of less than $0.07 \ mg \, Chl \, a \, m^{-3}$ at the surface. Based on this threshold and applying a constant conversion factor of $1.59 \ mg \, Chl \, a \, mmol \, N^{-1}$, we find that oligotrophic regions expand by 92% and 90% from 2000 to 2100 for OLIGO and GRAZ, respectively - both of which are in

the same order of magnitude as the 1998 to 2006 satellite based estimate of Polovina et al. (2008) (19% per decade).

Because of their underlying paradigm, OLIGO and GRAZ respond very differently to this increase in oligotrophic regions - even though they share a very similar behavior under historical conditions: in the simulation OLIGO, the diazotrophs can take advantage of the increasing vertical stratification because the diazotrophs are expertly exploiting oligotrophic conditions. The black line in Fig.3a shows that the biomass of diazotrophs increases globally along with an increase in size of the oligotrophic

regions. In addition, Fig.3b illustrates that, even though the size of oligotrophic regions and the diazotrophic biomass increases, the nitrogen fixation decreases in OLIGO. The reason is a reduced total supply of phosphorous to the surface in (expanded) oligotrophic regions which, in our model, puts an upper limit on nitrogen fixation. Additionally, nitrogen fixers have in the extended oligothrophic regions a competitive advantage over ordinary phytoplankton, because they require less P for their subsistence. Subsistence is, however, not necessarily correlated with high production rates which, ultimately, are limited by the

availability of phosphorous - whose limitation provided the niche for diazotrophs in OLIGO.

In contrast, the development of nitrogen fixation and abundance of diazotrophs goes much more hand-in-hand in GRAZ and the projected evolution of both variables differs strongly from OLIGO: an initial increase in both variables is followed by a subsequent decay. Such switching behavior at arbitrary tipping points (that are set by the model parameters) are typical for the rather generic zooplankton formulation used here and have been described, e.g., in Löptien and Dietze (2017), their Sect.3.2.

Note, however, that for both model versions the Bay of Bengal features a sudden pronounced onset of nitrogen fixation in the mid 2000s, which is visible even in the global mean. We anticipate that this abrupt change might be to a particularly pronounced sea surface temperature increase in this region.





# 4   Conclusions

We have been exploring two paradigms that are proposed in the literature to essentially define the environmental conditions
or niches where oceanic diazotrophs may thrive - even though they typically grow slower than ordinary phytoplankton, with
which they compete for essential resources such as phosphate. The first paradigm, a de-facto standard in global earth system
modelling, builds on selective grazing, where the diazotrophs are exposed to a grazing pressure that is reduced relative to
other, ordinary phytoplankton. The second paradigm, based on more recent considerations, assumes that diazotrophs have
lower (dissolved, particulate, organic and inorganic) phosphorous demands than other phytoplankton. Our experiments with
an Earth System Model of intermediate complexity show that both paradigms are apparently equally consistent with the most
recent and elaborate estimate of global oceanic nitrogen fixation, which combines observations with prognostic modelling and
data-assimilation techniques (Wang et al., 2019). Our experiments also show that the similarity of the model versions based
on the differing paradigms breaks in projections into a warming future. This suggests that current observational data are not
sufficient to constrain the sensitivity of the dynamics of bioavailable nitrogen under changing environmental conditions. We
find that for assessing the fidelity of respective paradigm additional observations of diazotrophic biomass are needed most. To
this end, the quota of nitrogen fixation versus the respective biomass of diazotrophs may prove especially helpful.

*Data availability.*  The model output is preliminary archived at https://nextcloud.ifg.uni-kiel.de/index.php/s/iYx7CWaGN8rx8KE and will
be provided in the Institute repository after review. The observational data collected by Luo et al. (2012) are available via
https://doi.pangaea.de/10.1594/PANGAEA.774851 and cited in the references.

*Author contributions.*  Both authors were involved in the design of the work, in data analysis, in data interpretation and in drafting the article.

*Competing interests.*  The authors declare that they have no conflict of interest.

*Acknowledgements.*  This work is a contribution of the project "Reduced Complexity Models" (supported by the Helmholtz Association of
German Research Centres (HGF) - grant no. ZT-I-0010) and funding from the German Research Foundation (DFG) under the grant no. LO
1377/3-1 ("Towards a deeper Understanding Cyanobacteria Blooms in the Baltic Sea"). This study benefited from DFG grant to support the
initiation of international collaboration "Reducing uncertainties in projected Southern Ocean Carbon fluxes" (LO 1377/5-1 und Di 1665/6-1).
We are grateful to Francois Primeau who provided the estimates on nitrogen fixation.



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





**Table 1.** Relevant model parameters in the reference model version REF and the setups GRAZ and OLIGO. Bold values refer to parameters obtained during tuning exercises. Note that $\theta_o$ has been calculated with the equation $\theta_d + \theta_o = 0.4$. The last column gives the ranges, in which the parameters are varied to minimize the misfit to the estimate by Wang et al. (2019).

| Param. | Description | Units | Ref. Value | GRAZ | OLIGO | Range |
|---|---|---|---|---|---|---|
| $C_d$ | Growth handycap of diazotrophs | unitless | **0.3** | 0.4 | 0.4 | 0.3-0.6 |
| $k_P^d$ | Half-saturation constant for $PO_4$-limitation for diazotrophs | mmol $PO_4$/m$^3$ | 0.044 | 0.044 | **0.009** | 0.002-0.044 |
| $\theta_d$ | Grazing preference for diazotrophs | unitless | 0.1 | **0.14** | **0.2** | 0.08-0.2 |
| $\theta_o$ | Grazing preference for phytoplankton | unitless | 0.3 | **0.26** | **0.2** | 0.2-0.32 |
| $J_O^{max}$ | Maximum potential growth rate of phytoplankton at $0^\circ C$ | day$^{-1}$ | 0.6 | 0.6 | 0.6 | - |
| $J_D^{max}$ | Maximum potential growth rate of diazotrophs at $0^\circ C$ | day$^{-1}$ | 0.6 | 0.6 | 0.6 | - |
| $k_{Fe}$ | Half-saturation constant for iron (Fe)-limitation | mmol Fe/m$^3$ | 0.1 | 0.1 | 0.1 | - |
| $k_N$ | Half-saturation constant for $NO_3$-limitation for phytoplankton | mmol $NO_3$/m$^3$ | 0.7 | 0.7 | 0.7 | - |
| $k_P$ | Half-saturation constant for $PO_4$-limitation for phytoplankton | mmol $PO_4$/m$^3$ | 0.044 | 0.044 | 0.044 | - |
| $\alpha$ | Initial slope of the P-I-curve | $(W/m^2)^{-1}d^{-1}$ | 0.1 | 0.1 | 0.1 | - |
| $T_b$ | E-folding temperature of biological rates | $^\circ C$ | 5.65 | 5.65 | 5.65 | - |



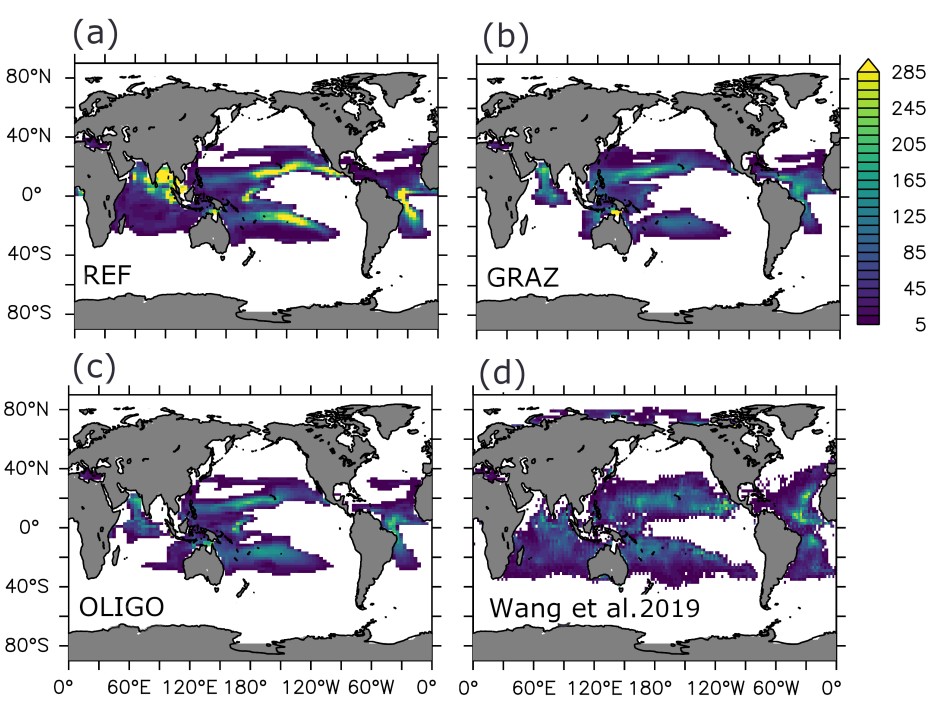

**Figure 1.** Simulated nitrogen fixation in $mmol\ N/m^2/yr$ for (a) the reference model, (b) the simulations based on OLIGO, (c) GRAZ and (d) estimate by Wang et al. (2019).



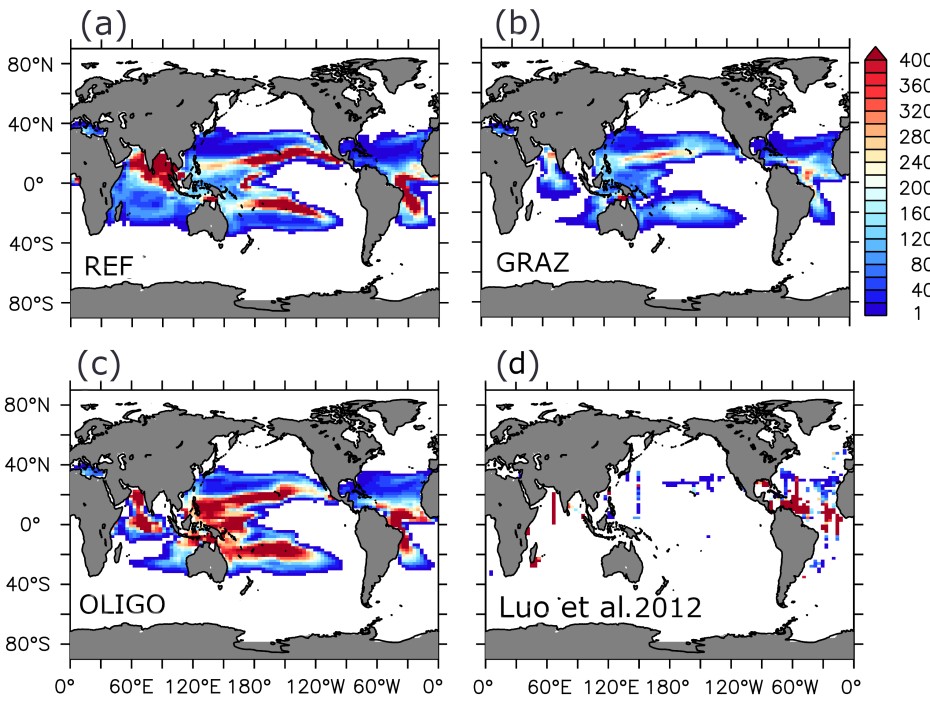

**Figure 2.** Annual mean non-zero diazotrophic biomass integrated over the upper 100m in $mg\ C/m^2$: (a) for the reference model, (b) the simulations based on OLIGO, (c) GRAZ and (d) as observed (Luo et al., 2012). The observations were interpolated linearly over depth and then gridded onto the model grid.

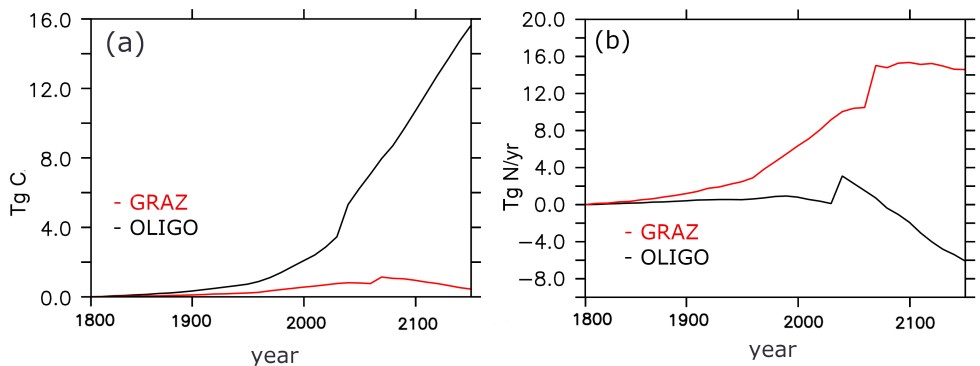

**Figure 3.** Anomalous projected evolution of (a) global annual mean biomass of diazotrophic biomass in units $Tg\ C$ and (b) global annual mean nitrogen fixation in units $Tg\ N\ yr^{-1}$.