# Peer review of "Contrasting juxtaposition of two paradigms for diazotrophy in an Earth System Model of intermediate complexity"

_Biogeosciences, 2020_

## Short Comment (SC1) · 11 Apr 2020

**Publisher's note: the content of this comment was adjusted on 2 April 2024 after approval of the BG co-editors-in-chief since some formulations were inappropriate.**

Comments by Andreas Oschlies, Chia-Te Chien, Wolfgang Koeve, Angela Landolfi, Markus Pahlow, Friederike Prowe and Christopher Somes

**General comments:**
The authors of this manuscript, Ulrike Löptien and Heiner Dietze, are colleagues of our research group. Because the manuscript of Löptien and Dietze (2020) addresses important scientific questions and to ensure that comments based on the knowledge built in our research group are used in a scientifically most productive way, we collectively decided to also respond publicly so that the authors and other interested readers may consider whether to take these into account in future work on this topic.

The manuscript discusses two interesting ideas, called paradigms by Löptien and Dietze (2020), on the controls of marine diazotrophs. It specifically contrasts frequently used earlier ideas on bottom-up controls with less prominent ones on top down controls by grazing. Unfortunately, the study fails to deliver on the goals set out in the abstract and introduction. It nevertheless calls for a consorted and perhaps overdue effort to examine the impact of grazing formulations on the simulated distribution and activity of diazotrophs, the importance of which was also indicated by the sensitivity experiments reported by Wang et al. (2019). Löptien and Dietze (2020) deserve credit for pointing this out clearly at the onset. Their approach of selecting passive switching by fixed grazing preferences of a linear grazing function can represent one first step (but note that the linear grazing function employed in eq.8 in Löptien and Dietze (2020) appears different from the Holling-type II grazing function used by Keller et al. (2012)). Major control is exerted also by the shape of the grazing function and by active switching (Prowe et al., 2012; Vallina et al., 2014). Some of the studies cited in the current contribution (Schmittner et al., 2008; Somes et al., 2010; Landolfi et al., 2013) use sigmoidal grazing functions. Other studies explicitly avoid different grazing preferences on diazotrophs and ordinary phytoplankton (Monteiro et al., 2010; Weber and Deutsch, 2014). Interestingly, the prognostic CESM model used in the Wang et al. (2019) study employs grazing preferences higher for diazotrophs than for diatoms, i.e. contrary to the results obtained for the GRAZ and REF experiments of Löptien and Dietze (2020). A more in-depth discussion of the treatment of grazing of diazotrophs in models is required, beyond qualitative statements like 'one assumption often implicitly incorporated . . . is selective grazing' (l.54-55).

The second paradigm of the study, called 'low-P demands' paradigm, also deserves a more careful investigation. There is, to our knowledge, no evidence for half-saturation constants for phosphate uptake being smaller in diazotrophs than in ordinary phytoplankton (e.g. Monteiro et al., 2010), as assumed as key mechanism in the OLIGO simulation. Line 60-63 states "The idea is that diazotrophs outcompete ordinary phytoplankton under P-depleted conditions because they can allocate more P to intracellular P-uptake machinery. The underlying physiological explanation is that diazotrophs presumably need less P to build their N-uptake machinery, which is already taken care of by the nitrogen-fixation machinery (which is known not to need so much P)." We are not aware of any observational evidence for this physiological explanation. Subsistence P (and also N) quotas estimated by Pahlow et al. (2013, Table 2) for *Trichodesmium* sp. were higher than for non-diazotrophic species and explained by the N and P requirements of the N2-fixation apparatus and consistent with a lower P use efficiency in diazotrophs during N2 fixation (Raven, 2012). It should be noted that the 'low-P demands' paradigm is not the mechanism that leads to the success of diazotrophs in P-depleted regions as investigated in different model settings (Pahlow et al., 2013, Landolfi et al., 2015), where the ability to access N via N2 fixation (and not P) allows diazotrophs to allocate more N (and not P) to acquire P.

An important aspect the authors neglect to discuss is feedbacks with denitrification, which can strongly impact patterns and rates of N2 fixation in the model and may respond differently under their different "paradigms", e.g. by different partitioning of newly fixed N into lateral and vertical export routes. In this respect it is important to know whether global NO3 and thus N2 fixation are in steady-state at the end of the tuning experiments run for 2000 years (line 151). Several studies have shown that a 2000yr spin up is short for global ocean tracers to reach steady state (e.g., Wunsch and Heimbach, 2008) and may be too short particularly for the nitrogen fixation and denitrification rates to reach equilibrium (Kriest and Oschlies, 2015).

The authors claim that their study is designed to constrain the envelope of model re-

sponses (l.69), but fail to provide an estimate of this constrained envelope. In particular the projections of diazotrophy under climate change presented at the end of the manuscript deserve estimates of uncertainty regarding their representativeness of the two 'paradigms'. Instead of focusing on differences in the sign-of-change for the two 'paradigms', the authors need to provide evidence that these differences are significant at the 'paradigm' level. Such an analysis could develop quite naturally from the parameter perturbation experiments carried out earlier in the manuscript. To prepare for such an analysis, the choice of model runs that aim to implement the two paradigms should be explained and justified in more detail. Implementation of the 'selective grazing' paradigm varies the growth penalty of diazotrophs and the relative grazing preferences on diazotrophs and ordinary phytoplankton. The tuned parameters reveal that grazing preferences in the GRAZ simulation differ less between diazotrophs and ordinary phytoplankton than they do in the REF run. Does this mean that REF is a more extreme case of the grazing paradigm than experiment GRAZ itself? Temperature dependencies of diazotrophs and ordinary phytoplankton remain different in all three configurations. With temperatures often correlating with surface nutrients, this may well map on the degree of OLIGO in all model configurations. It might help to illustrate how growth and grazing rates vary for a typical range of environmental conditions.

The conclusion that ''the development of a reliable model must be preceded by additional in-situ observations" (l.70) does not hold up to closer scrutiny and surely should not be interpreted as to stop model development until more or better data are available. In contrast, model development can actually point to those measurements that would be most valuable in reducing uncertainties. In particular, it remains unclear how the authors can justify the conclusion that more observations of biomass of diazotrophs are needed, rather than, for example, measurements of nitrogen fixation.

The conclusion that more observations are required before reliable models can be developed is even more difficult to understand when considering that the authors base their results entirely on the model-derived N2-fixation estimate of Wang et al. (2019)

instead of available observations of N2 fixations. Moreover, the Wang et al. (2019) estimate is based on a circulation model and partially inconsistent assumptions about elemental ecological stoichiometry. In particular, denitrification is, in the model of Wang et al. (2019) computed from modeled DON and DIN and observed (not modelled) oxygen concentrations. Wang et al. (2019) show that different model configurations they tested vary considerably in the simulated N2 fixation (their Fig. S4) and admit that the imperfect agreement with direct measurements of nitrogen fixation does not permit the identification of the one parameterization that is most realistic. Löptien and Dietze (2020) base their conclusions on the comparison with one model realization of Wang et al. (2019). It is important to know whether their conclusions would still hold if real data were used in the model assessments. Since its publication, the Luo et al. (2012) data base has been enhanced, particularly by adding new observations in the Pacific Ocean (e.g. Knapp et al., 2016; Landolfi et al., 2018).

The authors announce 'far reaching implications of both paradigms' in the context of climate change projections and the assessment of geoengineering options (l. 66ff), but fail to show implications of differing N2-fixation parameterizations on marine carbon uptake, warming projections, the efficacy of climate engineering options, or any related aspect in a quantitative manner. Given the small differences in projected N2-fixation (Fig. 3b ) for the two 'paradigms' (compared, e.g., with the uncertainty range of published N2-fixation estimates), more solid evidence needs to be provided to support the bold statement regarding 'far reaching implications'.

In summary, the manuscript fails to deliver on the goals set out in the abstract and introduction. It confirms that model configurations can be designed that show similar patterns for the present state of the ocean, but diverge under global warming (Löptien and Dietze, 2019). The manuscript raises several important issues, a careful investigation of which is desired and would be a scientifically very useful contribution to a better understanding of the controls of marine nitrogen fixation. The results are expected to deserve the additional work that is likely needed. In its current form, this manuscript

does not seem to have reached the degree of sophistication expected for a significant scientific step forward.

**Individual comments:**

l.6 the adjective phosphorous should be replaced by the noun phosphorus.

l.11 the call for more measurements of cyanobacteria biomass is not substantiated by the findings reported in the manuscript. Presumably, measurements of fixation rates would be much more beneficial.

l.14-15, The logic of this argument is difficult to follow. What matters for phytoplankton is what is dissolved in seawater rather than in the atmosphere. There is much more inorganic carbon dissolved in seawater than is N2.

l.21-22 Some reference for "Paleo-record suggest balanced by denitrification" would be useful here. The Altabet et al. (2002) study cited does not discuss the Holocene, perhaps Altabet (2007) would be better?

l.40. It is not so much the energetic cost of breaking the triple bond of N2 (which is very similar to that of reducing NO3 to NH4), but mostly the cost of getting rid of O2 that would otherwise destroy the enzyme nitrogenase (Grosskopf and LaRoche, 2012).

l.56. Selective grazing 'paradigm' need more careful description. The statement of 'implicitly incorporation' is not really true, see e.g. the early explicit discussion in Moore et al. (2001).

l.99 Presumably, this is a typo and units should be $\mu$mol m$^{-3}$ (also in Table 1).

l.102, eq. 2: there is an error in this equation. Presumably, the final term should read "max(0,(exp(T/Tb)-2.6))"

l.115, eq. 8. It should be mentioned that zooplankton also grazes on itself and on detritus, making the prey switching algorithm in the model a little more complex than suggested here. In addition, presenting the equation in this form does not make clear

that this is a Holling-type II functional response after Fasham et al. (1990). We unfortunately only now noticed that there was a typo in eq. 27 of Keller et al. (2012), which should read $\Theta = \phi_{P_O}P_O + \phi_{P_D}P_D + \ldots + K_{Graze}$ as it is coded correctly in the UVic 2.9 version of Keller et al. (2012).

l.122 The map of indirectly estimated N2 fixation should not be called 'observations'

l.127 it should be made clear that the term 'outperformance' used here is only valid with respect to a model-derived estimate of N2 fixation by Wang et al. (2019). In this regard, it would be helpful to clarify the very different model structure and parameterizations used in Wang et al. (2019), particularly with respect to stoichiometry and grazing formulations. When assessing the quality of the model, it would be better to use a model-data misfit as metric, i.e. observations of N2 fixation and/or observations of biogeochemical tracer distributions. It would be interesting to see how different the different model configurations behave with respect to real data.

l.132, eq. 9. This metric does not seem to be area/volume weighted. Is this OK?

l.136 Some more explanation would be useful as to why a second metric was introduced to exclude solutions with low biomass of diazotrophs. Would these still have yielded a good fit to the model-derived N2 fixation target?

l.147 Disregarding differences in the volume-weighted RMS errors of nutrient concentrations because they are smaller than the corresponding global standard deviations of the observations appears to rely on a very conservative criterion that does not necessarily rule out statistical significance of the differences. What would be the result of the same criterion applied to nitrogen fixation?

l.152. 'hindcasting historical conditions' is unclear. Presumably, what is meant is 'preindustrial'?

l.157 It is not clear why the data compiled by Luo et al. (2012) are too sparse to allow for comparison with observations. For example, Paulsen et al. (2017) or Dutkiewicz

et al. (2014; 2015) used the Luo et al. (2012) data base to calibrate their model, Monteiro et al. (2010) used direct measurements in their investigation of controls on global patterns of nitrogen fixation as well. While the spatial and temporal coverage of the data in Luo et al. (2012) is far from optimal, it may still serve as a useful constraint. Why not use model-data misfits of these direct measurements rather than resorting to an indirect model-derived estimate? What is the correlation between the Luo et al. (2012) (or its Landolfi et al. 2018 update with a lot more data added for the Pacific Ocean, including those obtained as part of the SFB 754 work at Kiel) with the Wang et al. (2019) estimate?

l.171 why not compare also with Luo et al. (2012) fixation rates?

l.173. "the simulated distribution of diazotrophs based on OLIGO is more realistic than in GRAZ. The biomass of diazotrophs in GRAZ is apparently too low." Given the very low optimized half-saturation constant for PO4 and the ability of model diazotrophs to take up NO3, the niche of diazotrophs may have expanded towards the niche otherwise occupied by oligotrophic non-fixing small phytoplankton. What is the percentage of NO3 uptake relative to the total N uptake in diazotrophs in OLIGO? And what is the contribution of diazotrophs to total biomass and total production in OLIGO?

l.177ff. An interesting result shown in Fig. 1 and Fig. 2 is that both OLIGO and GRAZ do not simulate any N2 fixation nor diazotrophs in the Bay of Bengal, which agrees much better with recent measurements (Löscher et al., 2020) than the notoriously high (and probably unrealistic) N2 fixation rates simulated by most current ESMs, including the REF simulation. Only at the end of the Results section do the authors say that both model formulations show an onset of N2 fixation in the Bay of Bengal in the middle of the 21st century (without showing any results or providing quantitative information). More information could turn this into a scientifically very useful result.

l.192-195 It would be interesting to understand why these ratios differ among the different models. Is this because of different surface nutrients, because of different grazing

pressures or because of different parameters of the diazotrophs?

l.200ff. This is an interesting result that the model is much more sensitive to parameter variations in the GRAZ configuration than in the OLIGO configuration (and presumably the REF configuration?). What is the reason for this different sensitivity? It would be helpful to make use of the results of the perturbed parameter simulations in Fig. 3 to get a more quantitative impression of the robustness of the results, e.g. by showing results of transient runs for all ensemble members that yield a cost function of some narrow range around the optimum of GRAZ and OLIGO (and REF?), respectively. A more detailed analysis might help to constrain the envelope of model responses, as claimed at the end of the introduction.

l.205 This is an interesting result that should be explored in more detail.

l.216 ff. This is an expected effect of global warming that has been discussed extensively in the literature, also with respect to its potential impact on biogeochemical cycles. References to respective papers should be included here to give proper credit.

**References:**

Altabet, M. A., M. J. Higginson, and D. W. Murray. The effect of millennial-scale changes in Arabian Sea denitrification on atmospheric CO2. Nature, 415, 159–162, 2002.

Altabet, M. A. Constraints on oceanic N balance/imbalance from sedimentary 15N records. Biogeosciences, 4, 75–86, 2007.

Dutkiewicz, S., B. A. Ward, J. R. Scott, and M. J. Follows. Understanding predicted shifts in diazotroph biogeography using resource competition theory. Biogeosciences, 11, 5445–5461, 2014.

Dutkiewicz, S., A. E. Hickman, O. Jahn, W. W. Gregg, C. B. Mouw, and M. J. Follows. Capturing optically important constituents and properties in a marine biogeochemical and ecosystem model. Biogeosciences, 12, 4447–4481, 2015.

Fasham, M. J. R., H. W. Ducklow, and S. M. McKelvie. A nitrogen-based model of plankton dynamics in the oceanic mixed layer. Journal of Marine Research, 48, 591–639, 1990.

Grosskopf T., and J. LaRoche. Direct and indirect costs of dinitrogen fixation in Crocosphaera watsonii WH8501 and possible implications for the nitrogen cycle. Frontiers in Microbiology, 3, 2012.

Keller, D. P., A. Oschlies, and M. Eby. A new marine ecosystem model for the University of Victoria earth system climate model. Geoscientific Model Development, 5, 1195–1220, 2012.

Knapp, A. N., K. L. Casciotti, W. M. Berelson, M. G. Prokopenko, and D. G. Capone. Low rates of nitrogen fixation in eastern tropical south pacific surface waters. Proceedings of the National Academy of Sciences, 113, 4398, 2016.

Kriest I., and A. Oschlies. MOPS-1.0: towards a model for the regulation of the global oceanic nitrogen budget by marine biogeochemical processes. Geoscientific Model Development, 8, 2929–2957, 2015.

Landolfi, A., H. Dietze, W. Koeve, and A. Oschlies. Overlooked runaway feedback in the marine nitrogen cycle: the vicious cycle. Biogeosciences, 10, 1351–1363, 2013.

Landolfi, A., P. Kähler, W. Koeve, and A. Oschlies. Global marine N2 fixation estimates: From observations to models. Frontiers in Microbiology, 9, 2112, 2018.

Löptien, U., and H. Dietze. Reciprocal bias compensation and ensuing uncertainties in model-based climate projections: Pelagic biogeochemistry versus ocean mixing. Biogeosciences, 16, 1865–1881, 2019.

Löptien, U., and H. Dietze. Contrasting juxtaposition of two paradigms for diazotrophy in an earth system model of intermediate complexity. Biogeosciences Discussions, 2020:1–16, 2020.

Löscher, C. R., W. Mohr, H. W. Bange, and D. E. Canfield. No nitrogen fixation in the Bay of Bengal? Biogeosciences, 17, 851–864, 2020.

Monteiro, F. M., M. J. Follows, and S. Dutkiewicz. Distribution of diverse nitrogen fixers in the global ocean. Global Biogeochemical Cycles, 24, GB3017, 2010.

Moore, J. K., S. C. Doney, J. A. Kleypas, D. M. Glover, and I. Y. Fung. An intermediate complexity marine ecosystem model for the global domain. Deep Sea Research Part II, 49, 403–462, 2001.

Paulsen, H., T. Ilyina, K. D. Six, and I. Stemmler. Incorporating a prognostic representation of marine nitrogen fixers into the global ocean biogeochemical model HAMOCC. Journal of Advances in Modeling Earth Systems, 9, 438–464, 2017.

Prowe, A. F, M. Pahlow, S. Dutkiewicz, M. Follows, and A. Oschlies. Top-down control of marine phytoplankton diversity in a global ecosystem model. Progress in Oceanography, 101, 1–13, 2012.

Raven, J. A. Protein turnover and plant RNA and phosphorus requirements in relation to nitrogen fixation. Plant Science, 188-189, 25–35, 2012.

Schmittner, A., A. Oschlies, X. Giraud, M. Eby, and H. Simmons. A global model of the marine ecosystem for long-term simulations: Sensitivity to ocean mixing, buoyancy forcing, particle sinking, and dissolved organic matter cycling. Global Biogeochemical Cycles, 19, 2005.

Somes, C. J., A. Schmittner, E. D. Galbraith, M. F. Lehmann, M. A. Altabet, J. P. Montoya, R. M. Letelier, A. C. Mix, A. Bourbonnais, and M. Eby. Simulating the global distribution of nitrogen isotopes in the ocean. Global Biogeochemical Cycles, 24, 2010.

Vallina, S. M., B. A. Ward, S. Dutkiewicz, and M. J. Follows. Maximal feeding with active prey-switching: A kill-the-winner functional response and its effect on global diversity and biogeography. Progress in Oceanography, 120, 93–109, 2014.

Weber T., and C. Deutsch. Local versus basin-scale limitation of marine nitrogen fixation. Proceedings of the National Academy of Sciences, 111, 8741–8746, 2014.

Wunsch C., and P. Heimbach. How long to oceanic tracer and proxy equilibrium? Quaternary Science Reviews, 27, 637–651, 2008.

---

## Referee Comment (RC1) · Anonymous Referee #1 · 21 May 2020

A premise of this paper is that two version of a model with different assumptions about diazotrophy and which have similar patterns of nitrogen fixation in the current day, have very different responses in a future climate scenario. This is a useful comment to make in terms of modeling of climate change impacts. The fact that different assumptions can lead to similar patterns of diazotrophy has indeed been seen before (e.g. Landolfi et al 2015). However, that future changes lead to different outcomes has not been documented to my knowledge. Given this premise, I would like to be supportive of this paper. However, there are several aspects that I have problems with, or that I think are too simplistic. And in the final assessment I am not convinced they have proved the premise. I am not convinced that these issues can be resolved. As such I do not

recommend publication.

1) The two paradigm concept is far too simplistic, and I believe unrealistic. See reviews by Sohm et al (2011) and more recent by Zehr and Carpone (2020), where much of the discussion of controlling mechanisms is focused on iron/phosphate availability perspective. In particular, the importance of iron is neglected in these paradigms and likely to be a more important than either grazing or phosphorus demands (see e.g. Ward et al 2013; Schlosser et al 2014). Moreover, I am not convinced that the "selective grazing" is a paradigm used by many models as stipulated (further references would be needed to show this to be true "de facto standard", line 247 needs substantiated). Early models of diazotrophy were based on Trichodesmium, which indeed appears to have lower grazing pressure and thus earlier models may have incorporated this type of parameterization. But it is now known that there is a great variety of diazotrophs (see e.g. Zehr et al 2020) and many do not appear to be grazed less than other phytoplankton. So this "paradigm" appears highly flawed. In fact, a study cited in this paper, Wang et al (2019) show a case where parameterizing reduced grazing on diazotrophs led to an unrealistic distribution of diazotrophs. It appears that such results are also found in this study (line 205-206). So why even make this a "paradigm"? Similarly, I am not convinced that the P-demand paradigm is fully justified. The study by Landolfi et al (2015) appears to have a very different parameterization of phosphate acquisition. It would seem that at least an iron paradigm should have been included (instead). Line 244: "...exploring two paradigms that are proposed in the literature" is too strong a statement. These do not appear to be the major paradigms that have been put forth (see reviews suggested above). Given the diversity of diazotrophs, it is likely that many processes lead to nitrogen fixation patterns, and expecting any single paradigm to explain them is simplistic. And as such, the setup of the paper appears fatally flawed.

2) Line 95: would have been better to be clearer what you mean by iron being not being explicitly resolved. Does that mean iron concentration are imposed? (I note that

it is not a state variable). Given that iron is likely important in controlling diazotroph distributions, this suggests in itself that this is not the best model for exploring controls on diazotrophy

3) All simulations (REF, OLIGO, GRAZ) have the assumption that diazotrophs do not grow above a temperature threshold and that they are dis-advantaged in nitrate-replete water (though this latter parameter is one that is explored, but only within a narrow range). Could it be that these two assumption are responsible for the similarities between the simulations in the current day model ocean. That is: any other assumptions (as in OLIGO or GRAZ) are slave to these other very strong restrictions. And then that it is the expansion of warmer, lower nitrate conditions in future change simulation that allow the two simulations to diverge. Put another way, these other assumptions (temp, NO3 handicap) are stronger controllers of the nitrogen fixation. So a "bad" parameterization is constrained by the temperature/handicap assumptions in such a way that it doesn't show up until warming occurs. This does not totally detract from the premise of the paper, but it does suggest that a "bad" parameterization could lead to unrealistic future projections. Which is an important difference to the premise.

4) The above also leads to the question on how reasonable the temp/handicap parameterizations are? There are cold water diazotrophs – as suggested by Harding et al (2018), by diazotrophy in places such as the Baltic Sea, and as shown in the Wang et al (2019) estimates shown in Fig 1d? A modelling study (Monteiro et al 2011) has shown that temperature does not need to be invoked to explain diazotroph distribtions. By constraining diazotrophy by temperature you have forced it to be close to observations, but not necessarily for the right reasons. How necessary is the NO3-replete handicap? I feel as though these two parameterizations should be far more fully understood before taking on this type of "paradigm" project.

5) Using Wang et al (2019) for the skill assessment seems awkward since Wang et al (2019) is itself a model estimate. Though data constrained, I would suggest it is not a good benchmark. Deutch et al (2007) was also a "data constrained" estimate

and it is very different to that found in Wang et al (2019). Wang et al (2019) is far more believable and a better study, but this example does suggest that there remains significant level of uncertainty even in a data constrained model,

6) Why not show the future diazotroph/nitrogen fixation distributions? Does GRAZ become unrealistic? I felt that since this was the crux of the premise, this last part of the paper was very rushed through: paragraphs only and one very simple figure. There is a mention of Bay of Biscayne feature (line 240), but this is not shown and appears rather arbitrary.

Details: Line 47: By "supply" do you mean "concentrations"? I would agree that concentrations do not necessarily correlate – but do not think that studies have shown the "supply" doesn't correlate as it is so difficult to measure supply rates. Line 89: Do you mean "DIC" not "DIN" Line 191: "desert dwellers" does not seem appropriate term here Line 193/256: The use of the word "quota" does not seem right here. Do you mean "ratio" instead? Line 194: I do not understand what you mean here? How do they go below zero? Line 218: why do discuss only phosphate here. The changes to nitrate are also important to the issue under discussion.

References: Deutch et al (2007). Spatial coupling of nitrogen inputs and losses in the ocean. Nature, 445, 163-167.

Harding et al., Symbiotic unicellular cyanobacteria fix nitrogen in the Arctic Ocean. Proc. Natl. Acad. Sci. U.S.A. 115, 13371–13375 (2018). doi: 10.1073/pnas.1813658115;

Landolfi, A., W. Koeve, H. Dietze, P. Kähler, and A. Oschlies (2015), A new perspective on environmental controls of marine nitrogen fixation, Geophys. Res. Lett., 42, 4482–4489, doi:10.1002/2015GL063756

Monteiro et al (2011), Biogeographical controls on marine nitrogen fixers, Global Biogeochem. Cycles, 25, GB2003.

Schlosser, C., et al. (2014), Seasonal ITCZ migration dynamically controls the location of the (sub)tropical Atlantic biogeochemical divide, PNAS, 111(4), 1438–1442, doi:10.1073/pnas.1318670111.

Sohm, J.A., E. A. Webb, D. G. Capone, Emerging patterns of marine nitrogen fixation. Nat. Rev. Microbiol. 9, 499–508 (2011). doi: 10.1038/nrmicro2594

Wang, W.-L., J. K. Moore, A. C. Martiny, F. W. Primeau, Convergent estimates of marine nitrogen fixation. Nature 566, 205–211 (2019). doi: 10.1038/s41586-019-0911-2
Ward, B. A., S. Dutkiewicz, M. Moore, and M. J. Follows (2013), Iron, phosphorus, and nitrogen supply ratios define the biogeography of nitrogen fixation, Limnol. Oceanogr., 58(6), 2059–2075, doi:10.4319/lo.2013.58.6.2059.

Zehr and Capone (2020). Changing perspectives in marine nitrogen fixation. Science, 368 eaay9514

---

## Referee Comment (RC2) · Anonymous Referee #2 · 25 May 2020

General Comments

The paper looks at two different formulations of Nitrogen fixation, which are then fitted to the nitrogen fixation observations to obtain the best solution. They show both formulations can adequately represent observations today but deviate when using the RCP8.5 future scenario. I like the inverse approach to parameterising the two N2-fixation formulations using observations. Interestingly, both formulations can represent today's limited N2-fixation data. However, to make the study more complete and justify publication, it needs expanding to address the following issues.

1. It is not clear what observations are used to constrain N2-fixation formulations. It is

stated that both models faithfully capture the other key BGC fields like NO3, PO4 and oxygen. However, you should show and quantify how well these fields are simulated by the best parameters of your two N2 fixation formulations. What are the differences? How about differences in DIC and air sea carbon fluxes, and volume of anoxic water? Do the differences provide any insight into the suitability of the 2 different formulations? No, can be the answer, but it would be helpful to show this more explicitly.

2. Typically in applying an inverse approach one considers other observations that were not used to constrain the model to assess the solutions. Here the future projected response is used, but you should consider other potential sources of information. A couple of ideas are: 1) does/would N15 differ between the two models?; 2) do any of the other BGC fields, like the ones listed in 1, differ significantly in the two formulations?; 3) does the ocean carbon uptake differ?; 4) does the response to ocean variability differ (e.g. ocean variability from atmospheric forcing of the last 5 decades)?; What I'm looking for is some guidance on whether other features of the two N2 simulations could provide useful insight to access their suitability and direct where to target future observations. Looking at natural variability in the ocean is one way to provide insight into how the two formulations respond in a way that could be assessed against our current understanding and observations. You should add this to the paper. I would also say that relying on more N2 fixation data would not enable one to choose the most suitable N2 formulation now since the simulated N2 fixation fields look similar. At what point in the future do the differences become significant? Is it the pattern or the total amount of N2 fixation that is most helpful in differentiating between the two formulations?

3. In the simulated future projection, the study only shows the global N2 fixation response of the two formulations, but do other BGC fields show significant differences too? How does the spatial distribution of N2 fixation change? Does an increase in N2 fixation significantly change ocean carbon uptake, equatorial net primary production, volume of anoxic water? Both the change in the amount and distribution of N2 fixation can impact the other BGC fields and fluxes in important ways - does this occur? I'm looking for reasons for why I should care about the future N2 fixation response? I assume the projected differences in the N2 fixation have impacts on the ocean BGC behaviour - it would be great if you showed it.

A few detail comments

line 15, nitrogen is also abundant in the ocean too line 19, not in the air but dissolved in the ocean line 22, what input? state it is the added Bio-available nitrogen line 31, not clear what is vicious cycles is - expand line 133 - only fit N2-fixation? how well do you simulated other BGC fields and fluxes?

line255 - observations show very low biomass of N2 fixers - is this believable? the two N2 formulations differ in the projected response of N2 fixation to global warming but could we use ocean variability over the past few decades to determine which one is more realistic?

---

## Author Comment (AC1) · 10 Jun 2020

The comment was uploaded in the form of a supplement:
https://www.biogeosciences-discuss.net/bg-2020-96/bg-2020-96-AC1-supplement.pdf

---

## Author Comment (AC2) · 10 Jun 2020

General Comments

**A: We thank the reviewer for the time and effort. We find all comments extremely helpful and are convinced that they will help us to improve our manuscript substantially.**

*R: The paper looks at two different formulations of Nitrogen fixation, which are then fitted to the nitrogen fixation observations to obtain the best solution. They show both formulations can adequately represent observations today but deviate when using the RCP8.5 future scenario. I like the inverse approach to parameterising the two N2-fixation formulations using observations. Interestingly, both formulations can represent today's limited N2-fixation data. However, to make the study more complete and justify publication, it needs expanding to address the following issues.*

*R: 1. It is not clear what observations are used to constrain N2-fixation formulations. It is stated that both models faithfully capture the other key BGC fields like NO3, PO4 and oxygen. However, you should show and quantify how well these fields are simulated by the best parameters of your two N2 fixation formulations. What are the differences? How about differences in DIC and air sea carbon fluxes, and volume of anoxic water? Do the differences provide any insight into the suitability of the 2 different formulations? No, can be the answer, but it would be helpful to show this more explicitly.*

**A: Agreed. We will add respective information to the revised version of the manuscript. We used estimates of nitrogen fixation to "tune" (poorly) known model parameters that are associated to the numerical representation of diazotrophy in the model. To first order this does have little effect on PO4, nitrogen and DIC because in our model, the stoichiometry (i.e., P:N:C) is identical for diazotrophs and ordinary phytoplankton, with the only exception being that diazotrophs are capable of "filling up" their intracellular N:P ratio up to Redfield even in the absence of NO3. Thus, NO3 is to a certain extend "produced" by diazotrophs (in contrast to ordinary phytoplankton). To this end, it seems straightforward to use global NO3 concentrations as a major constrain for the tuning of the diazotroph model formulations. The problem here is, however, that simulated anoxia and, in turn, denitrification rates are typically flawed as a consequence of an apparently endemic problem in the ocean circulation component of the current generation of global coupled ocean-circulation biogeochemical models (see Dietze & Löptien (2013), Getzlaff & Dietze (2013)). By fitting the diazotrophs to NO3 concentrations one risks to get the right answer for the wrong reason. We realize by the reviewer comment that this is**

**complex and will need a thorough discussion in the revised version of the manuscript.**

*2. Typically in applying an inverse approach one considers other observations that were not used to constrain the model to assess the solutions. Here the future projected response is used, but you should consider other potential sources of information. A couple of ideas are: 1) does/would N15 differ between the two models?; 2) do any of the other BGC fields, like the ones listed in 1, differ significantly in the two formulations?; 3) does the ocean carbon uptake differ?; 4) does the response to ocean variability differ (e.g. ocean variability from atmospheric forcing of the last 5 decades)?;What I'm looking for is some guidance on whether other features of the two N2 simulations could provide useful insight to access their suitability and direct where to target future observations. Looking at natural variability in the ocean is one way to provide insight into how the two formulations respond in a way that could be assessed against our current understanding and observations. You should add this to the paper. I would also say that relying on more N2 fixation data would not enable one to choose the most suitable N2 formulation now since the simulated N2 fixation fields look similar.At what point in the future do the differences become significant? Is it the pattern ort he total amount of N2 fixation that is most helpful in differentiating between the two formulations?*

A; **All these suggestions are very constructive. They make sense to us and are pretty straightforward to address. We will add a respective discussion and analysis to the revised version of the manuscript.**

*R: 3. In the simulated future projection, the study only shows the global N2 fixation response of the two formulations, but do other BGC fields show significant differences too? How does the spatial distribution of N2 fixation change? Does an increase in N2 fixation significantly change ocean carbon uptake, equatorial net primary production, volume of anoxic water? Both the change in the amount and distribution of N2 fixation can impact the other BGC fields and fluxes in important ways - does this occur? I'm looking for reasons for why I should care about the future N2 fixation response? I assume the projected differences in the N2 fixation have impacts on the ocean BGC behaviour - it would be great if you showed it.*

**A: One may indeed argue that, ultimately, the macronutrient PO4 (rather than the macronutrients NO3 and ammonium) is controlling the autotrophic growth -  or at least that is how the majority of state-of-the-art models used in coupled ocean-circulation biogeochemical model configurations are constructed. In a nutshell, most models have the following pattern engrained: in the presence of PO4 (and light, iron ...), diazotrophs grow if there is no or only little NO3 - otherwise ordinary phytoplankton outcompetes the diazotrophs. Thus, one might indeed argue that P-based models are sufficient to address most**

biogeochemical questions on a global scale: if there is PO4 (and light, iron ...) then autotrophic growth takes place, oxygen and organic matter are produced, organic matter is exported to depth where it's remineralization consumes oxygen etc. Among the reasons to consider NO3 nevertheless are: (1) academic curiosity over the question if the oceanic N-inventory is in balance (and if so on which timescales), brought up by Gruber and Sarmiento 1997, (2) the nitrogen cycle (including all sources and sinks) needs to be comprehensively understood if oceanic sources of the powerful greenhouse gas nitrous oxide are to be quantified, (3) management efforts to limit eutrophication in coastal region need to consider both, N and P, (4) blooms of nitrogen fixing cyanobacteria can be toxic which is an issue in some coastal areas where it can harm assets like tourism and fisheries, (4) a comprehensive understanding of diazotrophy is essential for non-constant Redfield ratio modelling which, in-turn, may well prove to be essential to reliable projections of biotic carbon uptake in a warming ocean (roughly speaking because the responses of C:P ratios in a warming world may differ between diazotrophs and ordinary phytoplankton).

However, we agree with the reviewer that we did not make our motivation entirely clear and also the manuscript would strongly benefit from an additional motivation. Following the reviewers, advice we started additional analysis. In accordance with our argumentation above we found that the respective projections are robust for many metrics. But there are exceptions. Among them a profound impact of the considered paradigms on the projected suboxic volume. We will include these new results into the revised version of the manuscript and thank the reviewer for the good suggestion.

*A few detail comments*

*R: line 15, nitrogen is also abundant in the ocean too*
A: True. We will clarify this.

*R: line 19, not in the air but dissolvedin the ocean*
A: Agreed. We will change that.

*R: line 22, what input? state it is the added Bioavailable nitrogen*
A: We will clarify this.

*R: line 31, not clear what is vicious cycles is - expand*
A: We will add a brief explanation.

*R: line 133 - only fit N2-fixation? how well do you simulated other BGC fields and fluxes?*

**A: As the chosen parameters refer specifically to diazotrophs, we tuned the model performance indeed based on observational estimates of nitrogen fixation only (because other oceanic state variables might depend crucially on other model parameters which are not "tuned"). We agree that such a choice contains inevitably to a subjective element.**

**That said, we will discuss the respective model performances also with respect to other BGC fields and fluxes in the revised version of the manuscript. The reviewer is correct in pointing out that this is important information.**

*R: line 255 - observations show very low biomass of N2 fixers - is this believable? the two N2 formulations differ in the projected response of N2 fixation to global warming but could we use ocean variability over the past few decades to determine which one is more realistic?*

**A: Good point. We will look into this and add a respective discussion to the revised version of the manuscript.**